# Genome Announcement of Four Parechovirus A3 Isolates from the United States of America

**DOI:** 10.3390/ijms26094378

**Published:** 2025-05-05

**Authors:** Debarpan Dhar, Terry Fei Fan Ng, Christopher J. Harrison, Eric Rhoden, Bernardo A. Mainou, Anjana Sasidharan, Katelyn E. VanDonge, Varun Chandra Boinpelly, Rangaraj Selvarangan

**Affiliations:** 1Department of Pathology and Laboratory Medicine, Emerging Infections, Children’s Mercy Research Institute, Kansas City, MO 64108, USA; ddhar@cmh.edu (D.D.); asasidharan@cmh.edu (A.S.); kevandonge@cmh.edu (K.E.V.); vboinpelly@cmh.edu (V.C.B.); 2Division of Viral Diseases, Centers for Disease Control and Prevention, Atlanta, GA 30333, USA; ylz9@cdc.gov (T.F.F.N.); imb0@cdc.gov (E.R.); qlk6@cdc.gov (B.A.M.); 3School of Medicine, University of Missouri, Kansas City, MO 64108, USA; harrisoncj@umkc.edu

**Keywords:** PeV-A3, whole genome sequencing, mutations, phylogeny

## Abstract

We report the complete genome sequences of four parechovirus-A3 (PeV-A3) isolates from Children’s Mercy Kansas City (CMKC), United States of America (USA): PeV-A3-MO-12-CMKC/CSF/MO/USA/2012 (isolated in 2012 from cerebrospinal fluid), PeV-A3-8C-CMKC/CSF/MO/USA/2022/ (isolated in 2022 from cerebrospinal fluid), PeV-A3-9C-CMKC/CSF/MO/USA/2022 (isolated in 2022 from cerebrospinal fluid), and PeV-A3-11B-CMKC/Blood/MO/USA/2022 (isolated in 2022 from blood). Sequence analysis revealed multiple mutations throughout the genome of the PeV-A3 isolates in comparison to the prototypic PeV-A3 A308/99 reference sequence (AB084913). Several unique amino acid changes were observed in the PeV-A3 isolates from 2022 that were absent in the PeV-A3 isolate from 2012. Phylogenetic analysis comparison determined that the sequenced PeV-A3 isolates from 2022 cluster together as a separate clade.

## 1. Introduction

The *Parechovirus* genus includes six species (Parechovirus A-F), with genotypes that infect humans belonging to the Parechovirus-A (PeV-A) species. To date, 19 PeV-A types have been identified [1]. PeV-A viruses have been detected in stool and respiratory specimens of healthy and ill (gastroenteritis and respiratory infections) patients, mostly children. PeV-A viruses have also been detected in blood and cerebrospinal fluid (CSF) specimens from patients, mostly in children and particularly infants less than 6 months of age, with severe illnesses (e.g., CNS infections or sepsis-like illness) [2,3]. The most frequently detected type is PeV-A1 [4], found most often in stool samples of infected patients. However, PeV-A3 is the type most frequently linked to severe illness and potential neurological symptoms or sequelae [5]. PeV-A3 was first reported in 2004 in Japan [6] and thereafter reported elsewhere globally [2].

PeV-A contains a single-stranded positive-sense RNA genome that is approximately 7300 nucleotides long and has a single open reading frame (ORF). The genome encodes a single large polyprotein, which is cleaved into three structural proteins (VP0, VP3, and VP1) and seven non-structural proteins (2A–C and 3A–D) [7]. Over time, drastic changes can occur across the PeV-A genome driven by the lack of proofreading activity of the PeV-A RNA polymerase, leading to a high mutation rate of approximately 2.5 × 10^−3^ substitutions per site per year [8]. Recombination has also been shown to occur between PeV-A3 and other PeV-A types resulting in changes across large stretches of the genome [9,10,11]. From an epidemiological standpoint, emergence of recombinant and/or mutated strains can lead to an increase in hospitalizations and clinical severity in infants [11,12]. Sanger-based sequencing of the VP1 capsid encoding gene and primers targeting the 5′untranslated region (UTR) have been the standard method for routine surveillance and genotyping of PeV-A [13,14]. However, this approach captures changes occurring only across a small region in the genome and hinders the understanding of changes across the entire viral genome. Analyzing whole genome sequences of PeV-A3 isolates will improve understanding of how PeV-A3 strains evolve and will be an important tool to predict changes in preparation for potential PeV-A3 outbreaks.

## 2. Results/Genome Announcement

Names of the sequenced viral isolates as well as their GenBank accession numbers, sequence length and closest identity to a reported sequence are provided in Table 1. A BLASTn analysis provided the highest alignment score for the PeV-A3-MO-12-CMKC/CSF/MO/USA/2012 isolate to the PeV-A3 isolate from Yamagata, Japan (LC864495.1). The PeV-A3-8C-CMKC/CSF/MO/USA/2022, PeV-A3-9C-CMKC/CSF/MO/USA/2022 and PeV-A3-11B-CMKC/Blood/MO/USA/2022 isolates aligned to another PeV-A3 isolate reported from Yamagata, Japan (LC864514.1). When compared with the prototypic sequence of PeV-A3-308/99 (AB084913), we identified multiple nucleotide changes spanning across the 5′ untranslated region (UTR) (Figure 1A) and 3′ UTR (Figure 1B) of the PeV-A3 isolates. Amino acid substitutions were also detected throughout the viral polyprotein (Figure 1C). Unique amino acid substitutions were observed in the PeV-A3 isolates from 2022 (PeV-A3-8C-CMKC/CSF/MO/USA/2022, PeV-A3-9C-CMKC/CSF/MO/USA/2022 and PeV-A3-11B-CMKC/Blood/MO/USA/2022) that were absent in the 2012 isolate (PeV-A3-MO-12-CMKC/CSF/MO/USA/2012). These substitutions were seen in non-structural proteins [2A (L776P, K781N, V832I, N877S, V885I, V915I); 2C (S1129A, I1178V, N1206D, A1311S, T1322I, K1344Q, Q1348K, Q1355E); 3A (S1418N, R1426K, E1440D, A1463V, V1466A, K1482R); 3B (A1498V, S1507T); 3C (I1577V, S1620N); 3D (I1714V, A1766S, S1811T, A1858T, Y1943F, S1958A, E1966D, D1976N, G2059N, I2060V)] and structural protein VP1 (R764G, A765V, A767V) (Figure 1C).

Phylogenetic analysis of whole genome sequences revealed that PeV-A3 isolate from 2012 (PeV-A3-MO-12-CMKC/CSF/MO/USA/2012) clustered together with isolates from Germany and Yamagata, Japan, that were collected in 2011 and 2012 (Figure 2, highlighted in red). The three isolates from 2022 (PeV-A3-8C-CMKC/CSF/MO/USA/2022, PeV-A3-9C-CMKC/CSF/MO/USA/2022 and PeV-A3-11B-CMKC/Blood/MO/USA/2022) are highly divergent and cluster as a distinct clade in the phylogenetic tree when compared with the closest available PeV-A3 genomes (Figure 2, highlighted in green). The closest genetic relatives of the 2022 isolates is to a PeV-A3 strain reported from Australia in 2019 (strain Q14f-37, MW076510.1) (Figure 2).

## 3. Discussion

PeV-A3 circulation in the United States follows a biennial pattern with peaks observed every alternate year [15]. Whole genome sequences of PeV-A3 isolates from outbreaks can provide genomic data on how PeV-A3 viral strains change over time. In this study, we sequenced the whole genome of four PeV-A3 clinical isolates from the United States, one from the 2012 outbreak (PeV-A3-MO-12-CMKC/CSF/MO/USA/2012) and three from 2022 (PeV-A3-8C-CMKC/CSF/MO/USA/2022, PeV-A3-9C-CMKC/CSF/MO/USA/2022 and PeV-A3-11B-CMKC/Blood/MO/USA/2022).

All the isolates have nucleotide changes in their 5′UTR and 3′UTR, plus amino acid substitutions in the viral polypeptide when compared to the genome of the prototypic PeV-A3-A308/99 strain (Figure 1A–C). The isolates from 2022 (PeV-A3-8C-CMKC/CSF/MO/USA/2022, PeV-A3-9C-CMKC/CSF/MO/USA/2022, PeV-A3-11B-CMKC/Blood/MO/USA/2022) have unique amino acid substitutions that were absent in the PeV-A3-MO-12-CMKC/CSF/MO/USA/2012 isolate and PeV-A3-A308/99 (Figure 1C). A higher propensity of C to U transitions relative to other mutations has been previously reported in picornaviruses [16]. Certainly, within our dataset of 188 PeV-A3 sequences used for multiple sequence alignments, we observed 378 sites of transitions and 119 sites of transversions. Within the transition sites, there were 120 sites with C to U transition and 105 sites that had a U to C transition. The implication of this transition is yet to be explored for PeV-A3.

While it is not surprising to detect PeV-A3-MO-12-CMKC/CSF/MO/USA/2012 as a close relative of other circulating PeV-As, the sequence divergence of 2022 PeV-A3 isolates (PeV-A3-8C-CMKC/CSF/MO/USA/2022, PeV-A3-9C-CMKC/CSF/MO/USA/2022, PeV-A3-11B-CMKC/Blood/MO/USA/2022) represented a new strain (<95% whole genome identity) that has not been characterized before and that is concurrent with their distinct phylogenetic position (Figure 2). Our method of phylogenetic analysis used a classical approach, assuming the generation of recombinant viruses from a recent common ancestor. Recent studies have provided validation of a phylogeny-free approach to better predict viral evolution [17,18], but the classical approach is still robust and computationally practical for a dataset like ours. Furthermore, we are aware that for many non-polio enteroviruses, routine Sanger sequencing is performed across the capsid region, particularly the VP1 region. Our phylogenetic tree has been constructed using a dataset of whole genome sequences and sequences >80% coverage to our input sequences, reported on GenBank. Therefore, these sequences used for analysis should have most of the 5′UTR and VP1 sequences included and so we do not envision any significant differences with a tree, if it would have been made exclusively with these specific regions.

Overall, our results highlight the importance of continuous molecular surveillance of emerging PeV-A strains. Ongoing research will assess the phenotypic impact of mutations on viral replication and pathogenesis.

## 4. Materials and Methods

Cells and Viruses: The Vero-Polio (Vero-P) cell line, as previously described [19], and four PeV-A3 isolates were provided by the Centers for Disease Control and Prevention, Atlanta, USA. Upon receipt, these viral isolates were passaged twice in Vero-P cells prior to nucleic acid extraction. Vero-P cells were maintained in Eagle’s Minimum Essential Medium (EMEM) (ATCC, Catalog #30-2003) supplemented with 10% Fetal Bovine Serum (Catalog #16140071, Thermo Fisher Scientific) and 1% Penicillin–Streptomycin (Catalog #15070063, Gibco).

Reverse transcription and random amplification of PeV-A3 nucleic acid isolates: Viral nucleic acids were extracted from PeV-A3 isolates using a QIAamp viral RNA minikit (Qiagen, Catalog # 52904) followed by DNase I treatment (rDNase I; Ambion of Thermo Fisher Scientific, Catalog# AM2222) following the manufacturer’s instructions. The extracted nucleic acids were amplified using a sequence-independent, single-primer amplification (SISPA) protocol that has been described previously [20]. Briefly, reverse transcription of viral RNA was performed using SuperScript III reverse transcriptase (Invitrogen, Thermo Fisher ScientificCatalog# 18080093) with a 28-base-pair reverse primer consisting of a 3′ end with eight random nucleotides (N1_8N, CCTTGAAGGCGGACTGTGAGNNNNNNNN). A complementary DNA (cDNA) strand was synthesized using the Klenow fragment of DNA polymerase I [(3′ to 5′exonuclease) (New England BioLabs, Catalog# M0209)]. Polymerase chain reaction (PCR) amplification was performed with AmpliTaq Gold polymerase (Thermo Fisher Scientific, Catalog# 4311806) and 100 mM of the forward primer in a 25 mL reaction volume using the following conditions: 1 cycle of 95 °C for 5 min, 5 cycles of 95 °C for 1 min, 59 °C for 1 min, and 72 °C for 90 s, followed by 25 cycles of 95 °C for 30 s, 59 °C for 30 s, and 72 °C for 90 s.

Library construction using NexteraXT and sequencing: Paired-end libraries from PeV-A3 isolates were generated using a NexteraXT DNA library preparation kit (Illumina, Catalog#FC-131-1024). Briefly, genomic DNA samples amplified from reverse transcription and random amplification were quantified on a Qubit High Sensitivity DNA kit (ThermoFisher Scientific, Catalog# Q32854) and visualized on a TapeStation (Agilent Technologies, Santa Clara, CA, USA). Thereafter, amplicons were tagged with adapter sequences using an Illumina DNA/RNA UD Indexes Set B, Tagmentation kit (Illumina, Catalog# 20091656) and then amplified to generate a library. The amplified libraries were cleaned using Illumina Purification beads (Illumina, Catalog# 20060057) and library quality was checked using an Agilent 2200 Tapestation system (Agilent Technologies, Santa Clara, CA, USA) using a High Sensitivity D1000 Screen Tape kit (Agilent Technologies, Catalog# 5067-5584, 5067-5585). Individual libraries were normalized and pooled to a final concentration of 1 nM. The pooled library was denatured and diluted, and 1.4 pM of denatured library was loaded for a 2 X 149 bp paired-end sequencing run on the MiniSeq platform (Illumina, Catalog# SY-420-1001) following the manufacturer’s protocol.

Data analysis: We analyzed the MiniSeq sequences by read mapping and gene annotation using Geneious 11.1.2 (Biomatters, https://www.geneious.com accessed on 2 December 2024), as described previously [20], using PeV-A3 genome references from GenBank. Identification of mutations across the genome was performed using SnapGene^®^ software, version 8.0.2 (from Dotmatics; available at snapgene.com). To obtain related genomes for subsequent phylogenetic analysis, standard nucleotide basic local alignment search using BLASTn [21] was performed on all four sequences against the NCBI non-redundant (nr) nucleotide database with default parameters. The output generated was 100 top hits per sequence. Redundant sequences were removed to eventually reduce the collection to a final of 188 PeV-A3 sequences. Sequences were aligned using MUSCLE [22] in MEGA 12 (Molecular Evolutionary Genetics Analysis) [23] with default parameters. The phylogenetic analysis was performed using the Neighbor-Joining (NJ) algorithm [24] to generate a phylogenetic tree with 1000 bootstrap replicates on MEGA12 (Figure 2).

## 5. Conclusions

In this manuscript, we report sequences of four PeV-A3 clinical isolates from the USA (PV290890-PV290893). Our data indicates that the 2022 clinical isolates of PeV-A3 (PeV-A3-8C-CMKC/CSF/MO/USA/2022, PeV-A3-9C-CMKC/CSF/MO/USA/2022, PeV-A3-11B-CMKC/Blood/MO/USA/2022) cluster as a separate clade and have unique mutations [2A (L776P, K781N, V832I, N877S, V885I, V915I); 2C (S1129A, I1178V, N1206D, A1311S, T1322I, K1344Q, Q1348K, Q1355E); 3A (S1418N, R1426K, E1440D, A1463V, V1466A, K1482R); 3B (A1498V, S1507T); 3C (I1577V, S1620N); 3D (I1714V, A1766S, S1811T, A1858T, Y1943F, S1958A, E1966D, D1976N, G2059N, I2060V); VP1 (R764G, A765V, A767V)] that were absent in the PeV-A3-MO-12-CMKC/CSF/MO/USA/2012 isolate.

## Figures and Tables

**Figure 1 ijms-26-04378-f001:**
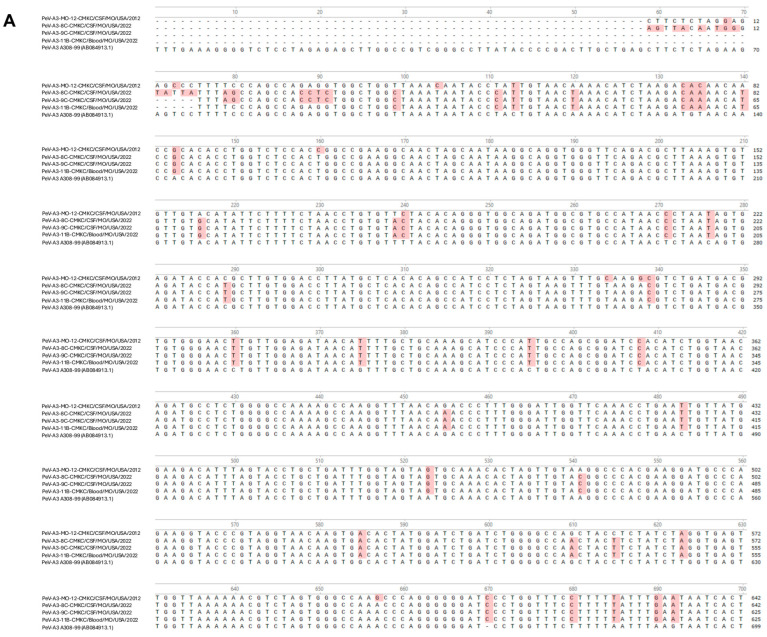
Nucleotide sequences of the (**A**) 5′ UTR and (**B**) 3′ UTR of the sequenced PeV-A3 isolates (PeV-A3-MO-12-CMKC/CSF/MO/USA/2012, PeV-A3-8C-CMKC/CSF/MO/USA/2022, PeV-A3-9C-CMKC/CSF/MO/USA/2022 and PeV-A3-11B-CMKC/Blood/MO/USA/2022) and prototypic PeV-A3 (PeV-A3-A308/99) reference sequence. Nucleotide changes in the sequenced isolates compared to the reference sequence are highlighted in red. (**C**) Amino acid sequences of the protein-coding region (VP0, VP3, VP1, 2A, 2B, 2C, 3A, 3B, 3C, 3D) of the sequenced PeV-A3 isolates (PeV-A3-MO-12-CMKC/CSF/MO/USA/2012, PeV-A3-8C-CMKC/CSF/MO/USA/2022, PeV-A3-9C-CMKC/CSF/MO/USA/2022 and PeV-A3-11B-CMKC/Blood/MO/USA/2022) and prototypic PeV-A3 (PeV-A3-A308/99) reference sequence. Unique amino acid substitutions in the PeV-A3 isolates from 2022 (PeV-A3-8C-CMKC/CSF/MO/USA/2022, PeV-A3-9C-CMKC/CSF/MO/USA/2022 and PeV-A3-11B-CMKC/Blood/MO/USA/2022) that were absent in the 2012 isolate (PeV-A3-MO-12-CMKC/CSF/MO/USA/2012) and PeV-A3-308/99 reference sequence are highlighted in red.

**Figure 2 ijms-26-04378-f002:**
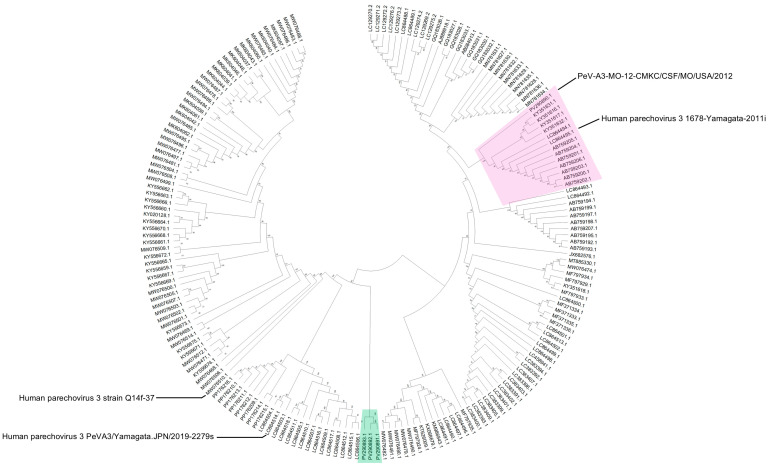
Phylogenetic analysis of whole genome sequences of four PeV-A3 isolates (PeV-A3-MO-12-CMKC/CSF/MO/USA/2012, PeV-A3-8C-CMKC/CSF/MO/USA/2022, PeV-A3-9C-CMKC/CSF/MO/USA/2022, PeV-A3-11B-CMKC/Blood/MO/USA/2022) with reported PeV-A3 sequences from GenBank. The tree was constructed using Neighbor-Joining method and 1000 bootstrap replicates on MEGA. The PeV-A3-MO-12-CMKC/CSF/MO/USA/2012 isolate (PV290890.1) clusters as a node with other reported PeV-A3 sequences (highlighted in pink). The post-pandemic 2022 PeV-A3 isolates (PeV-A3-8C-CMKC/CSF/MO/USA/2022: PV290891.1; PeV-A3-9C-CMKC/CSF/MO/USA/2022: PV290892.1; PeV-A3-11B-CMKC/Blood/MO/USA/2022: PV290893.1) cluster together as a separate node (highlighted in green). The bootstrap values are indicated on the branches.

**Table 1 ijms-26-04378-t001:** Table showing the names of sequenced viral isolates, their GenBank accession numbers, sequence length and the top alignment hit in BLASTn.

Isolate Name	Accession Number	Sequence Length (bp)	BLASTn Top Hit	
			Name and GenBank Accession Number	% Identity
PeV-A3-MO-12-CMKC/CSF/MO/USA/2012	PV290890	7264	Human parechovirus 3 1678-Yamagata-2011i (LC864495.1)	98.44
PeV-A3-8C-CMKC/CSF/MO/USA/2022	PV290891	7264	Human parechovirus 3 PeVA3/Yamagata.JPN/2019-2279s, (LC864514.1)	91.39
PeV-A3-9C-CMKC/CSF/MO/USA/2022	PV290892	7222	Human parechovirus 3 PeVA3/Yamagata.JPN/2019-2279s (LC864514.1)	91.47
PeV-A3-11B-CMKC/Blood/MO/USA/2022	PV290893	7222	Human parechovirus 3 PeVA3/Yamagata.JPN/2019-2279s (LC864514.1)	91.52

## Data Availability

The genome sequences of PeV-A3 isolates were deposited in GenBank under accession numbers PV290890-PV290893.

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
