# Peer review of "Genome Announcement of Four Parechovirus A3 Isolates from the United States of America"

_ijms, 2025, doi:10.3390/ijms26094378_

Round 1

Reviewer 1 Report

Comments and Suggestions for Authors

Comments to the authors on “Genome announcement of four parechovirus A3 isolates from the United States of America”.

This study reports the complete genome sequences of four parechovirus A3 (PeV-A3) isolates from the United States during 2012 and 2022. The isolates are collected from cerebrospinal fluid and blood samples. The authors identified multiple mutations in the 2022 isolates relative to the 2012 isolate and the prototypic PeV-A3-A308/99 strain, including unique amino acid substitutions in non-structural and structural proteins. Their phylogenetic analysis showed that the 2022 isolates clustered as a distinct clade, diverging from earlier strains and international sequences, suggesting the emergence of a novel PeV-A3 lineage. The findings are novel and highlight the importance of whole-genome sequencing for tracking viral evolution, predicting outbreaks, and understanding genomic changes. However, I have some suggestions that will make the study better.

Figure 1C can be revised to make it easy for readers to understand. For example, the protein-coding regions can be highlighted within the multi-color arrows for easy understanding of the figure without going to its legends.

Figure 2, phylogenetic analysis. Why are there no bootstrap values at the nodes? The authors can show bootstrap values higher than 80 or 90%, for example. Also, highlight the reference strains PeV-A3-308/99.

Author Response

For Communication article:

Genome announcement of four parechovirus A3 isolates from the United States of America

1.  Summary: We thank the reviewer for their thoughtful and constructive review of our manuscript. Your insights were very helpful in strengthening our results and improving the quality of our manuscript.

2. Questions for General Evaluation

Reviewer’s Evaluation

Response and Revisions

Does the introduction provide sufficient background and include all relevant references?

Yes

Are all the cited references relevant to the research?

Yes

Is the research design appropriate?

Yes

Are the methods adequately described?

Yes

Are the results clearly presented?

Yes

Are the conclusions supported by the results?

Yes

3. Point-by-point response to Comments and Suggestions for Authors

This study reports the complete genome sequences of four parechovirus A3 (PeV-A3) isolates from the United States during 2012 and 2022. The isolates are collected from cerebrospinal fluid and blood samples. The authors identified multiple mutations in the 2022 isolates relative to the 2012 isolate and the prototypic PeV-A3-A308/99 strain, including unique amino acid substitutions in non-structural and structural proteins. Their phylogenetic analysis showed that the 2022 isolates clustered as a distinct clade, diverging from earlier strains and international sequences, suggesting the emergence of a novel PeV-A3 lineage. The findings are novel and highlight the importance of whole-genome sequencing for tracking viral evolution, predicting outbreaks, and understanding genomic changes. However, I have some suggestions that will make the study better.

Comments 1: Figure 1C can be revised to make it easy for readers to understand. For example, the protein-coding regions can be highlighted within the multi-color arrows for easy understanding of the figure without going to its legends.

We appreciate the reviewer for this comment. We have redone Figure 1 to provide a better resolution and highlighted the regions of amino acid substitutions (Figure 1C) and nucleotide changes in the 5’ untranslated region (Figure 1A) and 3’ untranslated region (Figure 1B).

Comments 2: Figure 2, phylogenetic analysis. Why are there no bootstrap values at the nodes? The authors can show bootstrap values higher than 80 or 90%, for example. Also, highlight the reference strains PeV-A3-308/99.

Thank you for highlighting this point. We have also redone Figure 2 and marked the reference strains and provided bootstrap values. We have redone the multisequence alignment and phylogenetic tree using different software (MEGA12), described in materials and methods (Lines 158-170).

Reviewer 2 Report

Comments and Suggestions for Authors

This study reports a major evolutionary shift in Parechovirus isolates between 2012 and 2022. This major shift on the evolutionary tree is considered to be important data for examining the process by which new clades emerge from the effects of viral recombination, host immune pressure, population movements and changes, widespread vaccination, and improvements in public health on the circulation of Parechoviruses. Regarding the analysis process, general analysis methods for virus evolution are used accurately, and the materials, methods, and results are basically acceptable at present. 

However, for 5'UTR and 3'UTR, changes corresponding to the C->U transition should be analyzed as listed on “1:”. 

Additionally, in recent years, new analytical methods have been announced in this field, as shown below, and the Reviewer believes that it is necessary to discuss the possibility of bringing a new perspective to the emergence of the Parechovirus mutant strain reported by the author. Furthermore, the Reviewer strongly hopes that new predictions will be added and published by adding these analyzes to the data used in this study and the data that has been continuously obtained.

1. C->U transition: The relationship between 5'UTR and 3'UTR mutations and the C->U transition phenomenon of RNA viruses should be discussed. In this case, you should refer to “Extensive C->U transition biases in the genomes of a wide range of mammalian RNA viruses; potential associations with transcriptional mutations, damage- or host-mediated editing of viral RNA” by Peter SimmondsID*, M. Azim AnsariID.

2. The authors assume that only a single virus strain exists in a patient, but it has been reported that multiple strains exist in cases of viruses that mutate frequently and rearrange. 120 No. 5 e2219052120 - "It takes a village to build a virus" by Nash D. Rochman). Is this a possibility for Parechovirus? If so, it is possible to detect a mixture of strains by analyzing next-generation sequencing data (Ghorbani, A., Rostami, M., & Guzzi, P. H. (2024). Developing an AI-enabled pipeline for virus detection, validation, and SNP discovery from next-generation sequencing data.Frontiers in Genetics, 15, 1492752. doi:10.3389/fgene.2024.1492752.) should be added to Discussion.

3. For RNA viruses that mutate frequently, it is possible to accurately predict the process of virus strain development by assuming a gene network, rather than just assuming the generation of mutant strains from a common ancestor as in the current phylogenetic tree (Rochman, N. D. (2023). It takes a village to build a virus.Proceedings of the National Academy of Sciences, 120(5), Article e2219052120. https://doi.org/10.1073/pnas.2219052120). This possibility should be added in Discussion.

Author Response

For Communication article:

Genome announcement of four parechovirus A3 isolates from the United States of America

1. Summary

We thank the reviewer for their critical feedback. Some of the critiques helped us to think more about our data, that helped to strengthen the presentation of our findings.

2. Questions for General Evaluation

Reviewer’s Evaluation

Response and Revisions

Does the introduction provide sufficient background and include all relevant references?

Can be improved

We have added a few more references in the introduction section.

Are all the cited references relevant to the research?

Can be improved

We have removed some references and added new ones in the introduction and discussion section.

Is the research design appropriate?

Yes

Are the methods adequately described?

Yes

Are the results clearly presented?

Can be improved

We have provided better resolutions of our figures.

Are the conclusions supported by the results?

Yes

3. Point-by-point response to Comments and Suggestions for Authors

This study reports a major evolutionary shift in Parechovirus isolates between 2012 and 2022. This major shift on the evolutionary tree is considered to be important data for examining the process by which new clades emerge from the effects of viral recombination, host immune pressure, population movements and changes, widespread vaccination, and improvements in public health on the circulation of Parechoviruses. Regarding the analysis process, general analysis methods for virus evolution are used accurately, and the materials, methods, and results are basically acceptable at present. 

Comment 1: However, for 5'UTR and 3'UTR, changes corresponding to the C->U transition should be analyzed as listed on “1:”

Additionally, in recent years, new analytical methods have been announced in this field, as shown below, and the Reviewer believes that it is necessary to discuss the possibility of bringing a new perspective to the emergence of the Parechovirus mutant strain reported by the author. Furthermore, the Reviewer strongly hopes that new predictions will be added and published by adding these analyzes to the data used in this study and the data that has been continuously obtained.

1.    C->U transition: The relationship between 5'UTR and 3'UTR mutations and the C->U transition phenomenon of RNA viruses should be discussed. In this case, you should refer to “Extensive C->U transition biases in the genomes of a wide range of mammalian RNA viruses; potential associations with transcriptional mutations, damage- or host-mediated editing of viral RNA” by Peter SimmondsID*, M. Azim AnsariID.

We thank the reviewer for their valuable insight. We have analyzed our multisequence alignment and reported the number of transitions and transversions. Within that we have mentioned the number of CU transitions observed (Lines 101-105)

2.    The authors assume that only a single virus strain exists in a patient, but it has been reported that multiple strains exist in cases of viruses that mutate frequently and rearrange. 120 No. 5 e2219052120 - "It takes a village to build a virus" by Nash D. Rochman). Is this a possibility for Parechovirus? If so, it is possible to detect a mixture of strains by analyzing next-generation sequencing data (Ghorbani, A., Rostami, M., & Guzzi, P. H. (2024). Developing an AI-enabled pipeline for virus detection, validation, and SNP discovery from next-generation sequencing data.Frontiers in Genetics, 15, 1492752. doi:10.3389/fgene.2024.1492752.) should be added to Discussion.

We appreciate the reviewer for their insight. The viral isolates used for sequencing were originally isolated from sterile sites of the patients (cerebrospinal fluid). To our understanding, the possibility of two or more different PeV-A3 strains in a single sterile sites sample have yet to be described. So, while possible, two viruses simultaneously infecting blood or CSF seems unlikely.

3.    For RNA viruses that mutate frequently, it is possible to accurately predict the process of virus strain development by assuming a gene network, rather than just assuming the generation of mutant strains from a common ancestor as in the current phylogenetic tree (Rochman, N. D. (2023). It takes a village to build a virus.Proceedings of the National Academy of Sciences, 120(5), Article e2219052120. https://doi.org/10.1073/pnas.2219052120). This possibility should be added in Discussion.

We again thank the reviewer for this valuable suggestion. Such approach might be valuable for analyzing direct clinical samples that contain viruses with meaningful divergence. However, since this report investigated viral culture, this network approach probably is not suitable for this dataset (Lines 111-115).

Reviewer 3 Report

Comments and Suggestions for Authors

In this paper, the authors report the complete genome sequences of four parechovirus A3 strains isolated in 2012 and 2022. The results indicate that the three strains isolated in 2022, which form a distinct cluster in the tree, are new strains. This Reviewer would like to recommend publication of this paper after the authors address the following concerns.

  1. Bootstrap probabilities (%) should be indicated on some of the representative branches of the phylogenetic tree, and mention this issue in the Figure 2 legend.
  2. Provide a table with specific data (%) on the amino acid similarity between the three 2022-isolates and few strains with the shortest genetic distance on the phylogenetic tree (Fig. 2). In this case, mark the few strains on the phylogenetic tree (Fig. 2).
  3. Mark the four strains in Table 1 (Yamagata-related strains) on the phylogenetic tree (Fig. 2).
  4. Figure resolution is not sufficient. It is difficult to identify strain names in some of the figures. Provide figures with sufficient resolution.
  5. Many previous papers have reported virus genotypes based on 5' NCR and VP1 sequences, as described in this manuscript (lines 44-46). The authors should construct a phylogenetic tree based on the 5' NCR and VP1 sequences using same strains in Fig. 2, and discuss the similarities or differences of the topology between the figures. This data will be useful for surveillance work.

Author Response

For Communication article: Genome announcement of four parechovirus A3 isolates from the United States of America

1.     Summary: We appreciate the constructive critiques from the reviewer. The comments were addressed in our revised manuscript, helping us strengthen our results and analysis.

2. Questions for General Evaluation

Reviewer’s Evaluation

Response and Revisions

Does the introduction provide sufficient background and include all relevant references?

Yes

Are all the cited references relevant to the research?

Yes

Is the research design appropriate?

Yes

Are the methods adequately described?

Yes

Are the results clearly presented?

Must be improved

We have made new figures 1 and 2 providing better resolution and annotation of nucleotide/amino acid changes, and specific sequences in the phylogenetic tree.

Are the conclusions supported by the results?

Can be improved

Addressed in with questions 1-4.

3. Point-by-point response to Comments and Suggestions for Authors

In this paper, the authors report the complete genome sequences of four parechovirus A3 strains isolated in 2012 and 2022. The results indicate that the three strains isolated in 2022, which form a distinct cluster in the tree, are new strains. This Reviewer would like to recommend publication of this paper after the authors address the following concerns.

  1. Bootstrap probabilities (%) should be indicated on some of the representative branches of the phylogenetic tree and mention this issue in the Figure 2 legend.

Acknowledged and addressed in Figure 2.

  1. Provide a table with specific data (%) on the amino acid similarity between the three 2022-isolates and few strains with the shortest genetic distance on the phylogenetic tree (Fig. 2). In this case, mark the few strains on the phylogenetic tree (Fig. 2).

The closest genetic distance of the 2022 isolates is with a PeV-A3 strain reported from Australia in 2019. We have mentioned this in Lines 82-84 and highlighted the strain on the phylogenetic tree (Figure 2).

  1. Mark the four strains in Table 1 (Yamagata-related strains) on the phylogenetic tree (Fig. 2).

Acknowledged and addressed in Figure 2.

  1. Figure resolution is not sufficient. It is difficult to identify strain names in some of the figures. Provide figures with sufficient resolution.

We thank the reviewer for their valuable suggestion. We have now put new figures of nucleotide comparisons (Figure 1 A-C) and phylogenetic tree (Figure 2)

  1. Many previous papers have reported virus genotypes based on 5' NCR and VP1 sequences, as described in this manuscript (lines 44-46). The authors should construct a phylogenetic tree based on the 5' NCR and VP1 sequences using same strains in Fig. 2 and discuss the similarities or differences of the topology between the figures. This data will be useful for surveillance work.

We thank the reviewer for this suggestion. We are aware that for many non-polio enteroviruses, routine Sanger sequencing is performed across the capsid region, particularly the VP1 region. The phylogenetic tree has been constructed using whole genome sequences and sequences >80% coverage to our input sequences. Therefore, all these sequences used for analysis are inclusive of 5’UTR and VP1 and so we do not envision any significant differences in a tree made exclusively with these specific regions.

Round 2

Reviewer 3 Report

Comments and Suggestions for Authors

Overall, the authors have addressed the concerns raised by this Reviewer. Provide explanations in Discussion section regarding the tree topology, quoting the following response.

We thank the reviewer for this suggestion. We are aware that for many non-polio enteroviruses, routine Sanger sequencing is performed across the capsid region, particularly the VP1 region. The phylogenetic tree has been constructed using whole genome sequences and sequences >80% coverage to our input sequences. Therefore, all these sequences used for analysis are inclusive of 5’UTR and VP1 and so we do not envision any significant differences in a tree made exclusively with these specific regions.

Author Response

Acknowledged and addressed in Lines 115-121.